



# Biogeomorphic modeling to assess resilience of tidal marsh restoration to sea level rise and sediment supply

Olivier Gourgue[1,2], Jim van Belzen[3,1], Christian Schwarz[4], Wouter Vandenbruwaene[5], Joris Vanlede[5], Jean-Philippe Belliard[1], Sergio Fagherazzi[2], Tjeerd J. Bouma[3], Johan van de Koppel[3,6], Stijn Temmerman[1]

[1] Ecosystem Management Research Group, University of Antwerp, Antwerp, Belgium
[2] Department of Earth and Environment, Boston University, Boston, MA, United States of America
[3] Department of Estuarine and Delta Systems, NIOZ Royal Netherlands Institute for Sea Research, Yerseke, The Netherlands
[4] College of Earth, Ocean and Environment, University of Delaware, Lewes, DE, United States of America
[5] Flanders Hydraulics Research, Antwerp, Belgium
[6] Conservation Ecology Group, Groningen Institute for Evolutionary Life Sciences, University of Groningen, Groningen, The Netherlands

*Correspondence to*: Olivier Gourgue (ogourgue@gmail.com)

**Abstract.** There is an increasing demand for creation and restoration of tidal marshes around the world, as they provide
highly valued ecosystem services. Yet, tidal marshes are strongly vulnerable to factors such as sea level rise and declining sediment supply. How fast the restored ecosystem develops, how resilient it is to sea level rise, and how this can be steered by restoration design, are key questions that are typically challenging to assess. In this paper, we apply a biogeomorphic model to a planned tidal marsh restoration by dike breaching. Our modeling approach integrates tidal hydrodynamics, sediment transport and vegetation dynamics, accounting for relevant fine-scale flow-vegetation interactions (less than 1 m$^2$)
and their impact on vegetation and landform development at the landscape scale (several km$^2$) and on the long term (several decades). Our model performance is positively evaluated against observations of vegetation and geomorphic development in adjacent tidal marshes. Model scenarios demonstrate that the restored tidal marsh can keep pace with realistic rates of sea level rise and that its resilience is more sensitive to the availability of suspended sediments than to the rate of sea level rise. We further demonstrate that restoration design options can steer marsh resilience, as it affects the rates and spatial patterns of
biogeomorphic development. By varying the width of two dike breaches, which serve as tidal inlets to the restored marsh, we show that a larger difference in the width of the two inlets leads to more diversity in restored habitats. This study showcases that biogeomorphic modeling can support management choices in restoration design to optimize tidal marsh development towards sustainable restoration goals.

## 1 Introduction

Tidal marshes are among the most productive ecosystems on Earth (Barbier et al., 2011) providing invaluable services such as protection of coastal settlements against storms (Gedan et al., 2011; Zhu et al. 2020), carbon sequestration (Rogers et al.,



2019), maintenance of fisheries (Boesch and Turner, 1984) and water purification (Breaux et al., 1995). They are however among the most threatened ecosystems globally (Barbier et al., 2011). Over centuries, humans have built dikes to prevent tidal flooding and drained soils to gain land for agricultural, industrial, and urban expansion (Gedan et al., 2009). While
human-induced degradation and loss have accelerated in recent decades (Deegan et al., 2012; Wang et al., 2014; Tian et al., 2016), remaining tidal marshes are facing the additional global threat of accelerated sea level rise (SLR) caused by climate change (Spencer et al., 2016; Schuerch et al., 2018). In addition, the capacity of tidal marshes to adapt to SLR by sediment accretion and surface elevation gain can be compromised by decreasing sediment supply, for example due to upstream river dams and erosion control measures (Weston, 2014; Yang et al., 2020). Hence, efforts for conservation and restoration of
tidal marshes are increasing throughout the world (Mossman et al., 2012; Liu et al., 2016; Zhao et al., 2016; Waltham et al., 2021) with often as primary goal to support and rehabilitate biodiversity (Armitage et al., 2007; Weinstein, 2007) and provide nursery habitat for commercially important fish and invertebrate species (Rozas and Minello, 2001). Furthermore, marsh restoration is increasingly motivated by its role for nature-based shoreline protection, as marshes attenuate waves, currents and erosion and promote sediment accretion with SLR (Kirwan and Megonigal, 2013; Temmerman et al., 2013;
Barbier, 2014; Kirwan et al., 2016; Zhu et al., 2020) and for nature-based mitigation of climate change impacts through carbon sequestration (Barbier et al., 2011; Rogers et al., 2019). The success of restoration designs largely depends on the resulting rates of marsh vegetation development and sediment accretion, as they control the timescales at which target habitats, effective shoreline protection and carbon sequestration are reached. Besides, restoration designs must enable the development of marsh ecosystems that are resilient to modern threats such as SLR and decreasing sediment supply. Yet,
predicting actual rates of vegetation development and sediment accretion in establishing marshes, at timescales ranging from years to decades, remains to this day an important challenge (Fagherazzi et al., 2012; Mossman et al., 2012; Wiberg et al., 2020; Fagherazzi et al., 2020; Törnqvist et al., 2021).

Managed realignment, which consists in shifting the line of coastal defense structures landwards of their existing position, can create space for tidal marsh restoration or creation.  This practice has grown in popularity over the last two decades
(French, 2006; Turner et al., 2007), especially in the context of coastal squeeze and landward movement of the mean low water mark due to SLR and storms (Doody, 2013). Practically, a second line of defense is built landwards, while the first one is breached. The number and size of breaches are important design choices (Hood, 2014, 2015) and vary greatly between projects (e.g., Friess et al., 2014; Dale et al., 2017). As breaches become the inlets of the restored marshes, they have an important control on water and sediment volumes entering and leaving the system during each tidal cycle, and hence on
sediment accretion rates (Oosterlee et al., 2020). Other important design measures may involve excavating an initial channel network and treating soil conditions to facilitate soil drainage (O'Brien and Zedler, 2006), planting manually vegetation tussocks to ensure vegetation encroachment (Staver et al., 2020) or building hydraulic structures to control the tidal range and create optimal ecological conditions for vegetation development (Maris et al., 2007; Oosterlee et al., 2018). These design choices are mainly driven by restoration objectives and local environmental conditions. Yet, there is high uncertainty in how
restored tidal marshes develop. For example, several studies point at many restored sites that, in comparison with natural



tidal marshes, underperform in terms of biodiversity (Wolters et al., 2005; Mossman et al., 2012), topographic diversity (Lawrence et al., 2018), groundwater dynamics (Tempest et al., 2015; Van Putte et al., 2020) and biogeochemical functioning, including carbon sequestration (Santín et al., 2009; Suir et al., 2019). These outcomes can potentially hamper marsh ecosystem functions and the initial restoration objectives.

The rate at which tidal marshes develop in restoration projects is highly uncertain, yet so important. For example, sediment accretion rates determine whether restored tidal marshes can keep pace with local rates of SLR (Kirwan et al., 2010; Vandenbruwaene et al., 2011a; Webb et al., 2013; Kirwan et al., 2016). The establishment rate of pioneer vegetation and the succession towards climax vegetation may depend on small windows of opportunity that are very difficult to predict (Chambers et al., 2003; Hu et al., 2015; Cao et al., 2018). Furthermore, the rate of development is at the center of the tension

between public perception and restoration objectives. The public opinion is often very critical towards marsh restoration by managed realignment, as it implies the loss of valuable land, laboriously reclaimed by previous generations (Temmerman et al., 2013). On the one hand, fast development allows to quickly reach target habitats, which may support a positive public perception, but involves the risk of fast development towards a monotone climax ecosystem state. On the other hand, slow development (e.g., including bare mudflats) increases the risk of negative public perception in the first years, but may lead to

long-term persistence of high habitat diversity with different stages of succession. All these examples illustrate the need for modeling tools that are based on state-of-the-art scientific knowledge, and that allow to predict how fast restored tidal marshes develop and how development rates can be steered by restoration design.

Numerical models of tidal marsh development need to be able to simulate vegetation and landform evolution through feedbacks between hydrodynamics, sediment transport and vegetation dynamics. For example, sediment accretion rates

determine where and when the critical elevation, above which vegetation can grow in the intertidal zone, is reached (Bertness and Ellison, 1987; Balke et al., 2016; Bouma et al., 2016). The encroachment of vegetation in turn enhances sediment accretion through tidal flow deceleration, direct sediment trapping and organic matter accumulation (Vandenbruwaene et al., 2011b; Baustian et al., 2012; Fagherazzi et al., 2012). Sediment accretion processes are intrinsically linked to the availability of sediments at the marsh edge (Weston, 2014; Liu et al., 2021), as well as their tidal distribution

along complex channel networks and towards the vegetated platforms (Marani et al., 2003; Temmerman et al., 2005). At the same time, vegetation is known to control the morphology of channel networks and their efficiency to deliver sediments into the marsh interior (Kearney and Fagherazzi, 2016; Schwarz et al., 2018). So far, spatially explicit models that combine detailed hydro-morphodynamics with complex vegetation dynamics remain uncommon and their applicability is limited to relatively small domains (order of 1 km² or less) in comparison to real-life restoration projects of several km² (Temmerman

et al., 2007; Best et al., 2018; Schwarz et al., 2018; Bij de Vaate et al., 2020; Wang et al., 2021). In contrast, there are relatively fewer applications on large-scale domains (several km² or more) and these applications usually rely on relatively coarse grid resolutions (order of 100 m – e.g., Mariotti and Canestrelli, 2017), short simulation periods (order of 1 decade – e.g., Brückner et al., 2019), simplified hydro-morphodynamics (Craft et al., 2009; Alizad et al., 2016; Spencer et al., 2016; Mariotti, 2020; Mariotti et al., 2020) or simplified vegetation dynamics (D'Alpaos et al., 2007; Belliard et al., 2015). One of



the great challenges in numerical modeling of tidal marsh development is to combine large domains (order of 1 km$^2$ or more), fine grid resolution (order of 1 m$^2$ or less) and stochasticity in pioneer vegetation establishment, which are all essential to capture relevant scale-dependent biogeomorphic feedbacks, such as flow concentration between small-scale pioneer vegetation patches (order of m$^2$ – van Wesenbeeck et al., 2008; Gourgue et al., 2021) and its impact on the landscape-scale formation of channel networks (order of km$^2$ – Temmerman et al. 2007; Schwarz et al. 2018). To date, this

full range of scales in biogeomorphic feedbacks has never been included in a single numerical model for long-term tidal marsh development (from several decades to a century).

In this paper, we present a biogeomorphic model application to a specific tidal marsh restoration project by managed realignment, accounting for relevant fine-scale flow-vegetation interactions (less than 1 m$^2$) and their impact on vegetation and landform developments at the landscape scale (several km$^2$). The novelty of this paper is threefold. First, we evaluate the

long-term resilience (several decades) of a large-scale restored tidal marsh project (several km2) in response to different rates of SLR and different concentrations of suspended sediment supply. Second, we investigate how that resilience can be affected by restoration design options (here, the inlet configuration). Third, we evaluate our model performance against data on vegetation and geomorphic development in adjacent tidal marshes.

## 2    Methods

### 2.1    Multiscale biogeomorphic modeling


We have developed the biogeomorphic modeling framework Demeter to simulate explicitly, in an intertidal landscape, the feedbacks between (i) tidal hydrodynamics, (ii) sediment erosion, transport, deposition, and resulting bed level changes, and (iii) vegetation establishment, expansion and die-off for multiple dominant vegetation types. We adopt a multiscale approach (Fig. 1a), in which the hydro-morphodynamics is computed at a grid resolution of 5 m, while the vegetation dynamics is

computed at a much finer grid resolution of 25 cm. This multigrid approach enables to reduce the uncertainty arising from heterogeneous small-scale processes such as stochastic vegetation establishment (Temmerman et al., 2007; Schwarz et al., 2018) and plant-flow-sediment interactions around vegetation patches (van Wesenbeeck et al., 2008; Schwarz et al., 2015). It however requires the development of specific multiscale coupling techniques to preserve subgrid-scale heterogeneity while information is exchanged between the two modules (Gourgue et al., 2021). In this section, we describe the general principles

of our modeling approach. See Sect. S1 (supplementary material) for more details.





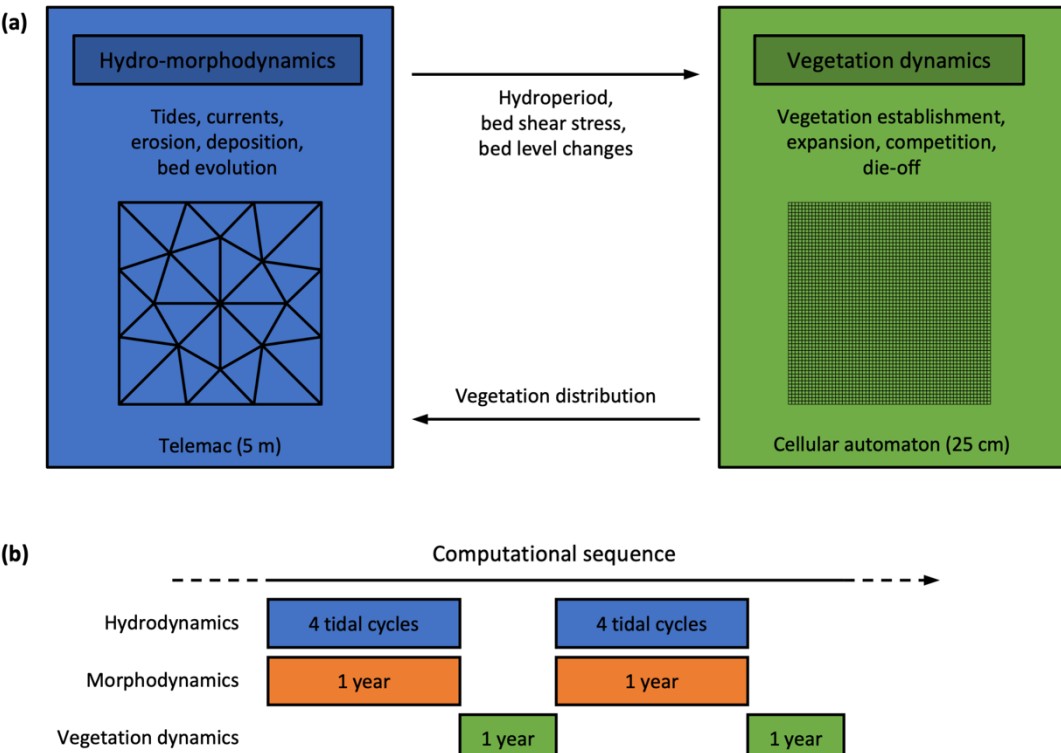

**Figure 1: Schematic representation of the multiscale biogeomorphic coupling. (a) Telemac is used for the hydro-morphodynamics at relatively coarse resolution (5 m) and a cellular automaton is used for the vegetation dynamics at finer resolution (25 cm). (b) Hydrodynamics and morphodynamics are constantly coupled (every time step). The morphological acceleration method allows to**
**upscale four semi-diurnal tidal cycles of hydrodynamics into one year of morphodynamic evolution. Hydro-morphodynamics and vegetation dynamics are coupled once a year.**

### 2.1.1 Hydro-morphodynamic module

Demeter is in principle compatible with any process-based hydro-morphodynamic model. Here we use the finite element solver suite Telemac (version 7.3.0), more specifically its modules Telemac-2D for hydrodynamics and Sisyphe for sediment
transport and morphodynamics. Telemac-2D simulates water level fluctuations and flow velocities by solving the depth-averaged shallow water equations in a two-dimensional horizontal framework (Hervouet, 2007). The vegetation resistance force is modeled as the drag force on a random array of potentially submerged rigid cylinders with uniform properties per species (Baptist et al., 2007) and calibrated against flume measurements (Vandenbruwaene et al., 2011b; Bouma et al., 2013; Gourgue et al., 2021). Sisyphe simulates the transport of cohesive sediments by solving the depth-averaged advection-
diffusion equation, as well as bed level changes through sediment erosion (Partheniades, 1965) and deposition (Einstein and Krone, 1962).





### 2.1.2    Vegetation module

Demeter includes a cellular automaton that is used here for the vegetation dynamics. A cellular automaton consists of a regular grid of cells, each one with a finite number of states (here, vegetation species). Cells can change their state in discrete
time steps, depending on their neighborhood state and a set of simple stochastic transition rules (Balzter et al., 1998). The cellular automaton simulates here the fate of three generic species representative of pioneer, middle marsh and high marsh plants encountered nearby the study site. Transition rules (i.e., for vegetation establishment, lateral expansion, species competition and vegetation die-off) are determined based on field observations and depend on environmental stressors provided by Telemac (i.e., hydroperiod, bed shear stress and bed level changes).

Based on observations in the vicinity of the study site, expected vegetation species that are representative for pioneer, middle marsh and high marsh vegetation are, respectively, *Aster tripolium*, *Scirpus maritimus* and *Phragmites australis*. In the model, the different species establish within different hydroperiod ranges. Within their hydroperiod range, *Aster tripolium* has a relatively high probability of establishment, but no possibility to expand laterally in patches, while *Scirpus maritimus* and *Phragmites australis* have relatively low probabilities of establishment, but their vegetated grid cells can expand
laterally and form patches that grow at rates of several m yr$^{-1}$. For each species, vegetated grid cells can potentially die if the bed shear stress or the erosion is too high. See Sect. S1.6 (supplementary material) for more details.

### 2.1.3    Biogeomorphic coupling

During one sequence of hydro-morphodynamics, Telemac keeps track of different environmental stressors (i.e., hydroperiod, bed shear stress and bed level changes), which are then used to regulate the transition rules for vegetation establishment,
lateral expansion, species competition and vegetation die-off in the vegetation module. During one sequence of the vegetation module, the cellular automaton updates the distribution of vegetation, which is then used to evaluate the vegetation resistance force in the hydro-morphodynamic module. See Sect. S1.3 and S1.4 (supplementary material) for more details.

### 2.1.4    Computation sequence

The computation sequence is schematized in Fig. 1b. We use the morphological acceleration method to decouple hydrodynamic and morphodynamic timescales (Lesser et al., 2004; Roelvink, 2006). The hydrodynamics is computed with time steps of 1.5 s. At every time step of the hydro-morphodynamic model, bed level changes are multiplied by the morphological acceleration factor $\alpha$, so that after simulating one semi-diurnal tidal cycle (12h25'), we have in fact extrapolated morphological changes to cover $\alpha$ cycles. The originality of our method is that we have selected four different
semi-diurnal tidal cycles (that is, one neap, one spring and two intermediate cycles, with high-water level intervals ranging from 0.3 to 0.5 m) from a larger, estuarine-scale hydrodynamic model. Each tidal cycle is considered representative of a class of tides with similar high tide levels (as computed from data records over 10 years in a station nearby the study site)



and is assigned with a different value of $\alpha$, which is determined proportionally to the relative number of tides in its own

class. The sum of all $\alpha$ values is determined so that the four representative tidal cycles correspond to one year of morphological changes. After one year of hydro-morphodynamics, the cellular automaton simulates one year of vegetation dynamics based on yearly aggregated information from the hydro-morphodynamic model (i.e., hydroperiod, bed shear stress and bed level changes). The updated spatial vegetation distribution is finally passed to the hydro-morphodynamic model, and as such the procedure is repeated to produce calculations of hydro-morphodynamics and vegetation dynamics for the next year. To mimic the effect of SLR, the bottom elevation is lowered every year by a value corresponding to the yearly increase

of mean sea level (MSL). All simulations in this paper are for a period of 50 years.

## 2.2 Study site: managed realignment of Hedwige-Prosper Polder

Hedwige-Prosper Polder (Fig. 2) is an agricultural area of 4.65 km² situated at the Belgian/Dutch border along the Scheldt Estuary. The surrounding region has a long embankment history, starting from the Middle Ages, including large scale intentional dike breaching for military operations in the late 16[th] century, and followed by subsequent stepwise re-

embankment projects until the early 20[th] century (Jongepier et al., 2015). In the coming years, the dikes of Hedwige Polder on the Dutch side (2.95 km²) and Prosper Polder on the Belgian side (1.70 km²) will be breached to re-introduce tides and restore natural intertidal habitats (Boerema et al., 2016). This managed realignment project is part of the Sigma Plan, which aims at improving flood protection in the tidal part of the Scheldt basin in Belgium (Smolders et al., 2015), while at the same time restoring natural estuarine habitat (Cox et al., 2006; Maris et al., 2007). Here, the tidal marsh restoration project consists

of building a new ring dike at the landward side of the area (Fig. 2c, green line) and removing parts of the old dike (Fig. 2c, red line) to allow daily tidal inundation and spontaneous development of an intertidal ecosystem. To facilitate the drainage of the area right from the start, the restoration design also includes the excavation of an initial channel network (Fig. 2c, blue line) and part of the fringing marshes and mudflats at the seaward side of the old dike (Fig. 2c, purple line). The excavated material is used to raise the landscape in Prosper Polder to partially compensate its lower elevation (Fig. 2c, yellow area) and

create breeding islands for birds (Fig. 2c, pink areas). By design, the restored tidal marsh will be a multi-inlet system, with a larger inlet in Hedwige Polder (further referred to as the Northern basin) and a smaller inlet in Prosper Polder (further referred to as the Southern basin). The restored area also includes the existing marsh of Sieperdaschor (Fig. 2c – further referred to as the old marsh). Right before de-embankment, the mean platform elevation with respect to the local mean high-water level (MHWL) will be -1.14 m in the Northern basin and -1.24 m in the Southern basin.

Local environmental conditions are determinant for the development of restored ecosystems (Liu et al., 2021). The Scheldt Estuary, here defined as the tidal part of the Scheldt River, is a semidiurnal macrotidal estuary extending over 160 km. At a gauge station near Bath (Fig. 2b), the tidal range has been recorded to vary on average from 4.21 m at neap tides to 5.76 m at spring tides during the period 2011-2015, and the MHWL to rise at a rate of 5.7 mm yr⁻¹ during the period 1931-2004 (Wang and Temmerman, 2013). This MHWL rise rate is used here as proxy for SLR rate (Sect. 2.3). The study site lies in the

brackish zone of the estuary, which is characterized by a steep gradient in salinity, with values ranging from 5 to 18 PSU



(Van Damme et al., 2005; Meire et al., 2005). Therefore, only a limited number of vegetation species (Sect. 2.1) can cope with the local environmental conditions. Finally, the local SSC is influenced by the presence of a maximum turbidity zone, where large volumes of cohesive sediments are concentrated and continually resuspended by the tidal flow (Baeyens et al., 1997; Chen et al., 2005; Meire et al., 2005). At the study site, the current average SSC is estimated at 63 mg l$^{-1}$ (Sect. S2, supplementary material).

Earth **Surface**
**Dynamics**
Discussions

EGU



**Figure 2:** Overview of the study site location within Northwestern Europe (a) and along the Scheldt estuary (b). Overview of the managed realignment project design (c) with close-up views of the small inlet according to the reference design (d) and three alternative designs (e-g), which consist of different breach sizes and the excavation of a straight channel between the dike breach and the main estuarine subtidal channel. The satellite image is a PlanetScope 4-band multispectral orthorectified scene (Planet Team, 2017) from July 4, 2019. The restoration design data are provided by the Vlaams-Nederlandse Scheldecommissie (VNSC). Other spatial data are from Natural Earth, Rijkswaterstaat, OpenStreetMap and EuroGeographics. © OpenStreetMap contributors 2020. Distributed under the Open Data Commons Open Database License (ODbL) v1.0.



### 2.3 Model scenarios

#### 2.3.1 Sea level rise rate and suspended sediment concentration

We investigate the resilience of the restored tidal marsh to human-induced climate and environmental changes by considering different relative SLR rates and different SSC at the seaward boundary (Table 1). If our model can account for changes in MSL (Sect. 2.1), changes in MHWL are more relevant for the biogeomorphology of tidal marshes, as the intertidal elevation relative to MHWL is determining the tidal inundation regime, hence affecting sediment accretion rates (Temmerman et al., 2004) and vegetation growth (Balke et al., 2016). Therefore, for the reference model scenario, we consider a SLR rate corresponding to the average rate of MHWL rise observed in the Scheldt Estuary over the last century, that is, 6 mm yr$^{-1}$ (Temmerman et al., 2004; Wang and Temmerman, 2013). This relatively high rate of MHWL rise is partly due to global SLR but also likely amplified by local human-induced changes in the geomorphology of the estuary, such as historical embankment of intertidal areas and widening and deepening of the navigation channels (Smolders et al., 2015; Wang et al., 2019). We also consider two additional scenarios, with no (0 mm yr$^{-1}$) and high (12 mm yr$^{-1}$) SLR rate, respectively.

Human activities can also potentially disturb the estuarine sediment dynamics. For example, river damming and erosion control measures in the upstream river catchment can potentially reduce the sediment supply to the estuary (Syvitski et al., 2009; Yang et al., 2020), while increasing tidal intrusion can potentially increase the SSC (Winterwerp et al., 2013). As sediment supply is critical for marsh development (Hopkinson et al., 2018; Liu et al., 2021), we consider three scenarios with different SSC at the seaward boundary, that is, the reference scenario with an SSC of 63 mg l$^{-1}$, as observed nearby the study site (Sect. S2, supplementary material), as well as two additional scenarios with low (30 mg l$^{-1}$) and high (120 mg l$^{-1}$) SSC, respectively.

#### 2.3.2 Small-inlet design

We also investigate the impact of restoration design on the rates of spatial vegetation and landform development. In particular, the width of the dike breaches (i.e., the inlets of the restored marsh) is an important design choice in managed realignment projects. Here, we simulate four different versions of the small inlet (Fig. 2d-g, Table 1). We first consider the reference design (Fig. 2d), which simply consists of a 50-meter-wide breach in the old dike. We then explore three alternative designs with respectively a 50-, 100- and 200-meter breach, along with the excavation of a straight channel of the same width between the dike breach and the main estuarine subtidal channel (Fig. 2e-g).

#### 2.3.3 Vegetation dynamics

In the reference model scenario, vegetation establishes randomly following different colonization strategies (i.e., either homogeneously with relatively high probability of establishment but no possibility to expand laterally, or patchily with relatively low probability of establishment but possibility to expand laterally to form growing patches – Sect. S1.2,





supplementary material) in areas where environmental stressors allow for it (Sect. 2.1.2). This is the expected behavior supported by field observations for the three selected species representative for pioneer, middle and high marsh vegetation (Sect. S1.5.2, supplementary material). To illustrate the impact of the vegetation dynamics on the biogeomorphic feedbacks and the model results, we also consider two variants of the reference model scenario. In the first variant, there is no vegetation. In the second variant, all species instantaneously colonize the entire areas for which environmental conditions are suitable.

**Table 1: Model scenarios.**

| Model scenario | Sea level rise rate | Suspended sediment concentration | Small inlet design | | |
| --- | --- | --- | --- | --- | --- |
| | | | Breach width | Excavated channel | Figure |
| #1 (reference) | 6 mm yr$^{-1}$ | 63 mg l$^{-1}$ | 50 m | No | Figure 2d |
| #2 | 0 mm yr$^{-1}$ | 63 mg l$^{-1}$ | 50 m | No | Figure 2d |
| #3 | 12 mm yr$^{-1}$ | 63 mg l$^{-1}$ | 50 m | No | Figure 2d |
| #4 | 6 mm yr$^{-1}$ | 30 mg l$^{-1}$ | 50 m | No | Figure 2d |
| #5 | 6 mm yr$^{-1}$ | 120 mg l$^{-1}$ | 50 m | No | Figure 2d |
| #6 | 6 mm yr$^{-1}$ | 63 mg l$^{-1}$ | 50 m | Yes | Figure 2e |
| #7 | 6 mm yr$^{-1}$ | 63 mg l$^{-1}$ | 100 m | Yes | Figure 2f |
| #8 | 6 mm yr$^{-1}$ | 63 mg l$^{-1}$ | 200 m | Yes | Figure 2g |

## 2.4    Evaluation of model performance

We have developed a model to predict the biogeomorphic evolution of a restored tidal marsh in an area that is still embanked. Due to lack of in situ data to validate our modeling approach, we evaluate its overall performance against observations in nearby intertidal mudflats and marshes. This is done for the reference model scenario (Sect. 2.3 and Table 1). Observations on elevation change (Sect. 2.4.1) and vegetation development (Sect. 2.4.2) are used for qualitative model calibration, based on model sensitivity to a selection of sediment parameters (i.e., critical shear stress for erosion, settling velocity, dry bulk density and incoming SSC) and vegetation parameters (i.e., establishment probability and patch expansion rate). Observations on channel network characteristics (Sect. 2.4.3) are used for qualitative model validation. In some cases, we calculate linear regressions from both model results and observations, and we perform an analysis of covariance (ANCOVA) to determine whether they are significantly different from each other (Sect. 2.4.4).



### 2.4.1    Sediment accretion on vegetated platforms

Sediment accretion processes on vegetated platforms are crucial for marsh resilience against SLR (Kirwan et al., 2016).
270   Based on digital elevation maps derived from historical topographic surveys in the adjacent marshes of the Drowned Land of
Saeftinghe (Fig. 2c) between 1931 and 1963 (Wang and Temmerman, 2013), we have developed an empirical relationship
between mean elevation change on vegetated platforms and mean high-water depth. Here, we develop a similar relationship
based on model results in the restored tidal marsh, using the same variables over the same time interval (i.e., between years
18 and 50 after de-embankment), and we compare it with the empirical relationship derived from observations. See Sect. S2
275   (supplementary material) for more details.

### 2.4.2    Pioneer vegetation development

The encroachment of vegetation on intertidal platforms impacts sedimentation in tidal marshes, and hence their geomorphic
development (Mudd et al., 2010). Here, we compare our model results with observed rate of spatial expansion of the
vegetation cover in the adjacent restored marshes of Paardenschor (Fig. 2c), from its de-embankment in 2004 until 2017. See
280   Sect. S3 (supplementary material) for more details.

### 2.4.3    Channel geometric properties

Channel networks control the flow of water and sediments in tidal marshes, and their evolution interact with the
biogeomorphic development of the surrounding intertidal platforms (D'Alpaos et al., 2007; Kearney and Fagherazzi, 2016).
Here, we compare various geometric properties of the simulated tidal channels with observations in the adjacent marshes of
285   the Drowned Land of Saeftinghe (Fig. 2c – Vandenbruwaene et al., 2013, 2015). To that end, we have developed a quasi-
automatic methodology to extract tidal channel networks and related geometric properties from model results. More
specifically, we compute the probability distribution of unchanneled flow length (i.e., the shortest distance to a channel
bank) as a measure of channel density (Tucker et al., 2001). The mean unchanneled flow length is calculated as the slope of
the linear portion of the probability distribution when plotted on semi-log axes (Marani et al., 2003; Chirol et al., 2018).
290   Along the channel network skeleton (i.e., the channel centerlines – Fagherazzi et al., 1999), we compute the watershed area,
the upstream mainstream channel length (i.e., the longest upstream channel within the watershed), the mean overmarsh tidal
prism (i.e., the mean high-tide water volume within the watershed for all tides overtopping the surrounding platform –
Vandenbruwaene et al., 2013, 2015) and the channel cross-sectional dimensions (i.e., channel width, channel depth and
channel cross-section area). We verify the applicability of Hack's law, an empirical power relationship that links watershed
295   area and mainstream channel length (Rigon et al., 1996). We also verify the applicability of O'Brien's law, an empirical
relationship that links tidal prism and channel cross-section area (D'Alpaos et al., 2009, 2010). See Sect. S4 (supplementary
material) for more details.



### 2.4.4    Linear regressions and analysis of covariance

In several cases, we calculate linear regressions from both model results and observations. First, we split the data into 10 sub-samples of equal size, based on the x-axis variable (i.e., the first sub-sample includes the 10% smallest values, etc.). Then, we calculate the mean value of each sub-sample, and we derive the linear regressions from these mean values. Finally, we perform an ANCOVA to determine whether linear regressions from model results and observations are significantly different from each other. Two null hypotheses are tested. The first null hypothesis is that the slopes of the regression lines are equal, and the second that their intercepts are equal. For each of these two tests, we calculate the associated p-value. Two regression lines are significantly different from each other if at least one p-value is lower than 0.05. If both p-values are higher than 0.05, the regression lines are statistically equal.

### 2.5    Vertical datums and metrics used for result analysis

Elevation data are either expressed with reference to the Normaal Amsterdams Peil (NAP), the official vertical datum in the Netherlands, which represents approximately the MSL at the Dutch coast, or the local MHWL.

The mean platform elevation over a certain area (i.e., the Northern basin, the Southern basin, or both combined) is calculated excluding channels and supratidal areas (i.e., the highest parts of the dikes and constructed bird breeding islands). The vegetation cover over a certain area is calculated as the proportion of vegetated cells, excluding supratidal areas.

The ebbing time is a proxy for the drainage efficiency of an inlet and is calculated as the time needed to drain 95% of the total ebb volume. The ebb volume is the instantaneous volume of water flowing through an inlet since high tide during the ebb phase (i.e., from high to low tide in the estuary). The total ebb volume is the ebb volume at the end of the ebb phase (i.e., at low tide). The relative ebb volume is the instantaneous ebb volume divided by the total ebb volume.

## 3    Results

### 3.1    Reference model scenario

The results of the reference model scenario indicate that the system is predominantly depositional (Fig. 3f-j), with only localized erosion up to about 10 cm yr$^{-1}$ concentrated near the inlets and in some initially excavated channels in the first ten years (Fig. 3f) and in some newly formed channels afterwards (Fig. 3g-j). In general, the development of vegetation (Fig. 3k-o) follows the topographic evolution (Fig. 3a-e), as higher elevations decrease hydroperiods and increase establishment chances. Replicate simulations of the reference model scenario indicate that stochasticity in vegetation development influences where secondary channels form.

In the first 10 to 20 years, most sedimentation occurs in the back of the Northern basin and almost everywhere in the Southern basin (Fig. 3a-b and Fig. 3f-g), where pioneer vegetation and then middle marsh vegetation start to encroach whenever the elevation, and hence the hydroperiod, becomes suitable (Fig. 3k-l). The middle marsh vegetation ability to



colonize its surroundings via clonal expansion (Sect. 2.1) is especially visible in years 20 and 30 (Fig. 3l-m). In general, middle marsh and high marsh vegetation follow similar colonization strategies, but with a 10- to 20-year delay (Fig. 3l-o).

In the first 20 to 30 years, some of the initially excavated channels are predicted to fill up (Fig. 3f-h) or even to disappear over time (Fig. 3c-e), especially those oriented perpendicular to the tidal flow propagation. Meanwhile, newly formed channels are primarily created by differential deposition (Fig. 3g-h), although erosion starts to occur in a second phase when they become narrower (Fig. 3h-j). In general, vegetation does not encroach in tidal channels before they fill up, because the hydroperiod and the shear stress are too high.

Overall, the presence of vegetation has a positive impact on platform accretion rates in the Northern basin, although the speed of colonization has nearly no influence on the mean platform elevation 50 years after de-embankment (Fig. S1a, supplementary material). In the Southern basin, neither the presence of vegetation nor the speed of colonization seems to affect sediment accretion on the platforms (Fig. S1b, supplementary material), which suggests that the hydrodynamics is predominant in that part of the restored marsh. Locally, the vegetated dynamics can have remarkable geomorphic effects,

such as the maintenance or disappearance of pre-excavated channels, whether we consider no vegetation, the reference vegetation dynamics, or instantaneous colonization (Fig. S2).

Earth **Surface**
Dynamics
Discussions



**Figure 3: Reference model scenario (#1). Bed elevation (a-e), bed elevation change (f-j) and vegetation distribution (k-o) every 10 years after de-embankment. The bed elevation change is calculated as the mean difference over the previous 10 years. The dashed lines delineate the old marsh, the Northern basin, and the Southern basin. All figures are rotated by 43° clockwise, as compared to Fig. 2c.**




### 3.2 Evaluation of model performance

Our results for the reference model scenario are in good agreement with the empirical relationship between mean elevation change on vegetated platforms and mean high-water depth, which is obtained from observations in a tidal marsh close to the

study site (Fig. S3, supplementary material). The linear regressions obtained from model results and observations are statistically equal, as both their slopes ($p = 0.496$) and their intercepts ($p = 0.412$) are not significantly different (Table S1, supplementary material). This suggests that sediment accretion processes on vegetated platforms are well represented in our model. Predicted vegetation cover development is also in good agreement with observations in another restored tidal marsh close to the study site (Fig. S4, supplementary material).

The predicted channel network 50 years after de-embankment is slightly less dense, as compared with observations in a nearby tidal marsh (Fig. 4a), with mean unchanneled flow lengths of respectively 26.0 m (model) and 23.7 m (observations). The exponent of Hack's law is 0.908 for the model results and 0.909 for the observations (Fig. 4b). These values are statistically equal as the slopes of the linear regressions are not significantly different ($p = 0.913$ – Table S1, supplementary material). However, their intercepts are significantly different ($p = 0.007$), which means that the predicted channel lengths

are smaller than those observed in the nearby natural marsh, with regards to the local watershed area. The exponent of O'Brien's law is 0.71 for the model results and 0.87 for the observations (Fig. 4e). These values are not statistically equal as the slopes of the linear regressions are slightly, but significantly different ($p = 0.023$). The intercepts are also significantly different ($p = 0.004$), which means that predicted channel cross-section areas are larger than those observed in the nearby tidal marsh, with regards to the local tidal prism. The relatively important deviations between model results and observations

in terms of channel width and channel depth (Fig. 4c-d and Table S1, supplementary material) partly compensate each other, so that the discrepancy in channel cross-section area is much smaller, but also decreases with increasing tidal prism (Fig. 4e). This suggests that appropriate volumes of water, and hence suspended sediments, are conveyed through the channel network and towards the intertidal platforms.

Earth **Surface**
**Dynamics**
Discussions

EGU



**Figure 4: Reference model scenario (#1).** Channel geometric properties 50 years after de-embankment (blue) compared to observations in an established marsh close to the study site (black). Probability distribution of the unchanneled flow length (a), upstream mainstream length vs. watershed area (b), and channel width (c), channel depth (d) and channel cross-section area (e) vs. mean overmarsh tidal prism. (b-e) Model results and observations are respectively split into 10 sub-samples of equal size (Sect. 2.4.4). Markers and error bars represent the geometric means and standard deviations of each sub-sample, respectively. Dashed lines represent geometric regressions of the geometric means.

### 3.3 Impact of sea level rise and sediment supply

Our results indicate that the restored tidal marsh is resilient to realistic ranges of SLR rate and SSC at the seaward boundary (Fig. 5). Indeed, for every model scenario considered, the intertidal platforms accrete faster than SLR (Fig. 5a-b) and the restored area can sustain a growing vegetation cover (Fig. 5c-d).

In Fig. 5a-b, we show the mean platform elevation with respect to MHWL, which increases over time due to SLR, rather than a fixed level like NAP. This is especially relevant when comparing the system response to different SLR rate scenarios. In terms of absolute platform elevation (i.e., with respect to a fixed level like NAP), increasing rates of SLR leads to faster platform accretions. However, the rate at which intertidal platforms catch up with MHWL decreases with increasing rates of




Earth **Surface**
**Dynamics**
Discussions

SLR (Fig. 5a). This is especially relevant in terms of marsh resilience, which is defined as the ability to keep pace with SLR.

Platform elevation (Fig. 5a-b) is also related to vegetation development (Fig. 5c-d), which is driven by hydroperiod rather than absolute elevation. That is why we choose to present the mean platform elevation with respect to the MHWL.

By the end of the 50-year simulated period, the system has not reached an equilibrium, where both mean platform elevation and vegetation cover converge to a stable value. Nevertheless, it can be extrapolated from our results that equilibrium will occur slower for increasing rate of SLR (Fig. 5a, c) and decreasing sediment supply (Fig. 5b, d). Interestingly, within the

range of considered parameter values, the rate of biogeomorphic development seems more sensitive to suspended sediment availability (Fig. 5b, d) than SLR rate (Fig. 5a, c).

**Figure 5: Sea level rise rate (a, c) and suspended sediment concentration (b, d) model scenarios (#1-5). Evolution of the mean platform elevation with respect to the mean high-water level (MHWL) (a-b) and development of the vegetation cover (c-d) in the**

**Northern and Southern basins combined.**





### 3.4    Impact of inlet width

To investigate the impact of inlet width, we first analyze separately how the large-inlet Northern basin and the small-inlet Southern basin (Fig. 2c) behave for the reference scenario. We then compare model scenarios with increasing inlet widths in the Southern basin.

Our results indicate that, for the reference design, the Southern basin develops faster than the Northern basin (Fig. 6a-b). During the flood phase (Fig. 7a-c), water and suspended sediments fill the Southern basin through the small inlet, but also from the Northern basin. Conversely, in the first part of the ebb phase (Fig. 7d-e), water and suspended sediments leave the Southern basin both through the small inlet and towards the Northern basin. However, in the second part of the ebb phase (Fig. 7f), the water surface elevation in the Southern basin drops below the Northern basin platforms, so that water and
suspended sediments can only flow towards the small inlet. Meanwhile, the small inlet slows down the evacuation of the remaining water, giving more time to suspended sediments to settle in the Southern basin.

This interpretation is confirmed by an analysis of the water volume and sediment mass budgets during a spring tide in the first year after de-embankment (Fig. 8). At the boundary between the Northern and Southern basins, there is 40% more water entering the Northern basin during the flood phase than leaving it during the ebb phase (Fig. 8a, pink). Consequently, the
small-inlet ebb discharge is 40% higher than its flood discharge (Fig. 8a, blue). Besides, it takes 40 more minutes to flush the water through the small inlet than through the large inlet (Fig. 6c), which gives 30% more time for suspended sediments to settle in the Southern basin after high tide. This leads to 60% more sedimentation in the Southern basin than in the Northern basin (Fig. 8b, black). The same pattern can be observed, albeit to a lesser degree, for intermediate and neap tides, as well as at later stages of marsh development.

Our argument that faster biogeomorphic development in the Southern basin is due to enhanced sediment trapping related to its small inlet size is further supported by the three alternative design scenarios (Fig. 2e-g). Our results indeed demonstrate that widening the small inlet slows down the biogeomorphic development in the Southern basin (Fig. 9b, d). It has however the opposite effect in the Northern basin, where widening the small inlet leads to faster biogeomorphic development (Fig. 9a, c). At the landscape scale, widening the small inlet leads to the formation of more, larger channels that originate from the
small inlet and expand into the Northern basin (Fig. 10, ellipses). This contributes to the supply of more sediments into the Northern basin. Interestingly, while the width of the small inlet clearly affects the spatial distribution of sedimentation and vegetation development in the restored marsh, it barely impacts the general biogeomorphic trend when the Northern and Southern basins are considered combined (Fig. S5, supplementary material). Overall, it is a zero-sum game where the Northern and Southern basins act like communicating vessels.

The ebbing time remains very close to an average value of 3h08' (standard deviation of 2') for both inlets and all designs (Fig. 9e-f), except in the case of the small inlet in its reference design (3h54').



Earth **Surface**
**Dynamics**
Discussions

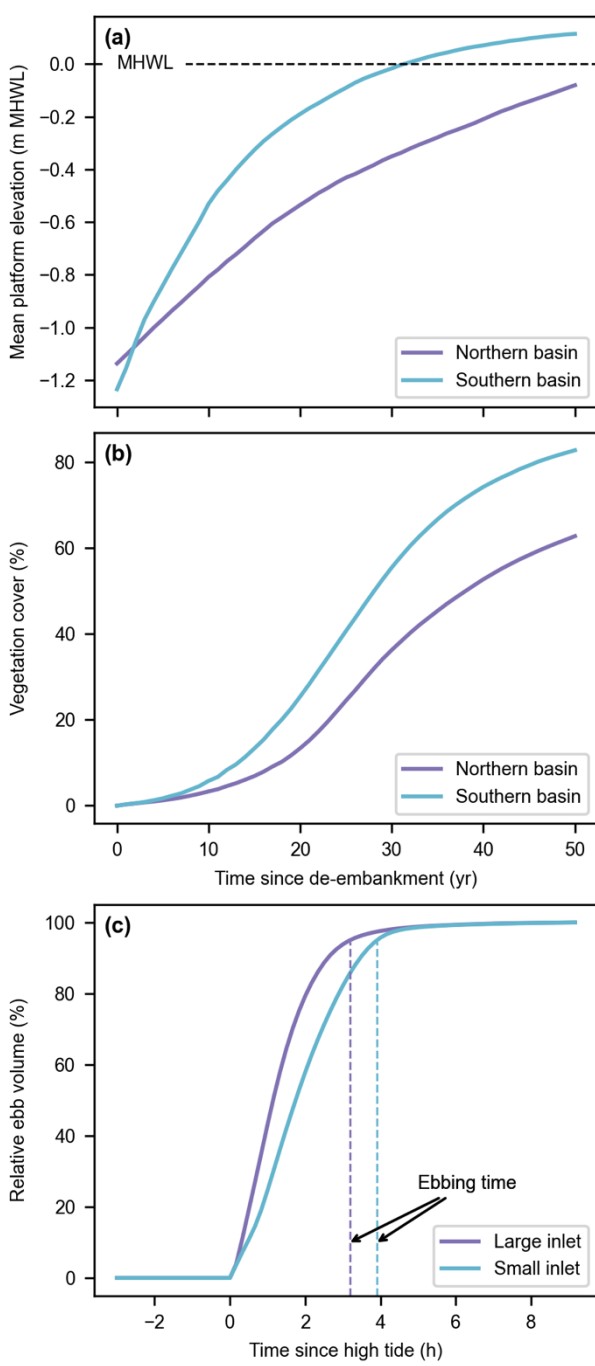

**Figure 6: Reference model scenario (#1). Evolution of the mean platform elevation with respect to the mean high-water level (MHWL) (a) and development of the vegetation cover (b) in the Northern and Southern basins. Evolution of the relative ebb volume (i.e., the percentage of water volume flowing through an inlet since high tide during the ebb phase, see Sect. 2.5) and resulting ebbing time (i.e., the time corresponding to a relative ebb volume of 95%, proxy for inlet flush efficiency, see Sect. 2.5) for the large and small inlets during a spring tide of the first year after de-embankment (c).**


Earth **Surface**
**Dynamics**
Discussions



**Figure 7: Reference model scenario (#1). Water depth (color maps) and flow velocity (grey arrows) every 50 minutes during a**
**spring tide of the first year after de-embankment. Time tags are relative to high tide. Flow velocity arrows are not displayed if**
**their magnitude is lower than 0.1 m s$^{-1}$ or if the water depth is lower than 10 cm. The dashed lines delineate the old marsh, the**
**Northern basin, and the Southern basin. All figures are rotated by 43° clockwise, as compared to Fig. 2c.**



Earth **Surface**
Dynamics
Discussions



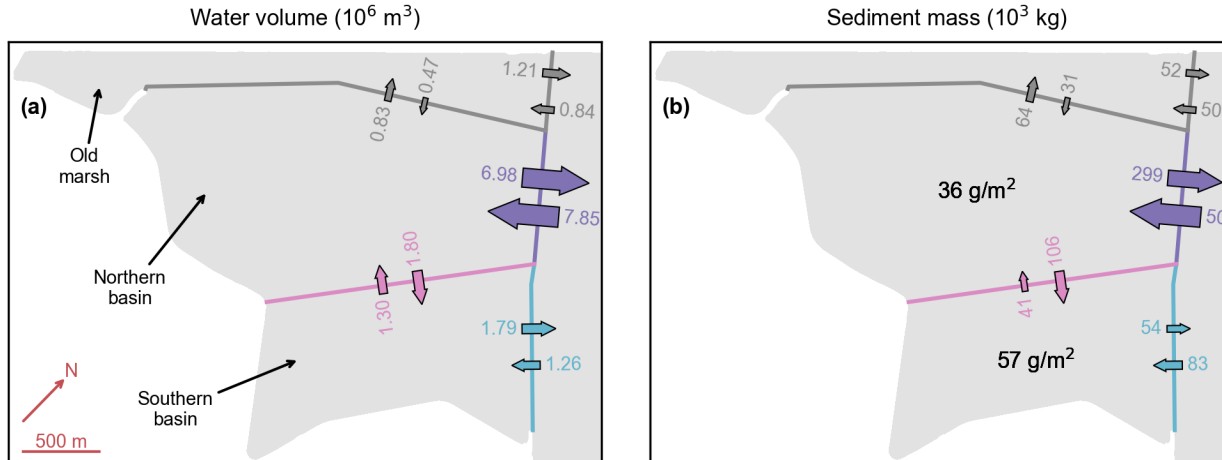

**Figure 8: Reference model scenario (#1). Water volume (a) and sediment mass (b) budgets during a spring tide in the first year after de-embankment. Color numbers and arrows represent water volumes and sediment masses flowing into and away from the different basins. Black numbers represent the resulting sediment deposition during that period. All figures are rotated by 43° clockwise, as compared to Fig. 2c.**





**Figure 9: Inlet design model scenarios (i.e., reference design and three alternative designs with small-inlet widths of respectively
50, 100 and 200 m, and an excavated channel – #1, 6-8). Evolution of the mean platform elevation with respect to the mean high-
water level (MHWL) (a-b) and development of the vegetation cover (c-d) in the Northern (a, c) and Southern basins (b, d).
Evolution of the relative ebb volume (i.e., the percentage of water volume flowing through an inlet since high tide during the ebb
phase, see Sect. 2.5) and resulting ebbing time (i.e., the time corresponding to a relative ebb volume of 95%, proxy for inlet flush
efficiency, see Sect. 2.5) for the large (e) and small (f) inlets during a spring tide of the first year after de-embankment.**



**Figure 10: Inlet design model scenarios (i.e., reference design (a-e) and three alternative designs with small-inlet widths of respectively 50 m (f-j), 100 m (k-o) and 200 m (p-t), and an excavated channel – #1, 6-8). Bed elevation every 10 years after de-embankment. The dashed lines delineate the old marsh, the Northern basin, and the Southern basin. The ellipses emphasize the channel network that originates from the small inlet and its expansion into the Northern basin with increasing small-inlet size. All figures are rotated by 43° clockwise, as compared to Fig. 2c.**





## 4 Discussions

The increasing demand for creation and restoration of resilient tidal marshes calls for the development of new modeling tools. These models must be able to assess whether rates of biogeomorphic development will achieve restoration objectives, as well as how these rates can be eventually steered by restoration design. In this paper, we present a novel biogeomorphic
model application to a tidal marsh restoration project in a macrotidal estuary. We find that the ability of the restored marsh to sustain future rates of SLR is more sensitive to suspended sediment availability than to rates of SLR (Fig. 5). We also demonstrate that inlet design can steer rates and spatial patterns of biogeomorphic development (Fig. 9-10), and as such can be used to achieve specific restoration objectives.

### 4.1 Biogeomorphic modeling as a tool to optimize tidal marsh restoration design

One of the main objectives in every tidal marsh restoration project is for intertidal platforms to build vertically faster than SLR, allowing to develop and maintain a vegetation cover. In that context, local environmental conditions play a crucial role. In the case studied here, our model predicts that, for the first 50 years, the restored tidal marsh can keep pace with realistic rates of SLR, and that its resilience is more sensitive to suspended sediment availability (Fig. 5). These findings are in line with previous studies on marsh adaptability to SLR. For example, an ensemble model study indicates that tidal marshes with
similar sediment input and tidal range as our study site can cope with SLR rates up to 70 mm yr$^{-1}$ in the 21st century (Kirwan et al., 2010, 2016). Here, we test SLR rates up to 12 mm yr$^{-1}$ in the first 50 years after de-embankment, for which we predict a highly resilient restored marsh. Furthermore, a recent global assessment of tidal marsh and mangrove restoration projects reveals that the ability of coastal wetlands to keep pace with SLR is above all driven by suspended sediment availability (Liu et al., 2021), which is also in line with our results.

A second key question that is typically raised when planning for tidal marsh restoration is how fast the ecosystem and its different habitat zones develop (Yando et al., 2019), which is for a large part driven by the rates of sediment accretion, pioneer vegetation establishment and succession. Here, the desired rates are depending on the restoration objectives. High rates of sediment accretion and vegetation development allow to quickly reach certain restoration objectives, such as different aspects of nature-based shoreline protection. For example, high-lying and densely vegetated marshes are most
effective for wave attenuation (Möller, 2006; Schoutens et al., 2020; Willemsen et al., 2020), and reduction of shoreline erosion (Möller et al., 2014; Francalanci et al., 2013; Wang et al., 2017; Schoutens et al., 2019) and dike breaching hazards (Zhu et al., 2020). However, when tidal marshes are created along estuaries or deltas to attenuate extreme high tides and storm surges (Smolders et al., 2015; Stark et al., 2017; Huguet et al., 2018; Smolders et al., 2020), lower accretion rates are preferred to maintain higher water storage capacity in the restored tidal marshes. When the objective is to restore intertidal
habitats and meet certain biodiversity goals (Hinkle and Mitsch, 2005; Chang et al., 2016), it may be favorable for accretion rates to be high so that vegetation can develop fast, but also not too fast so that a diversity of habitats can persist over time, including tidal channels, mud flats, pioneer marsh and higher marsh vegetation, while avoiding a rapid succession to climax




species. In the case studied here, our model predicts that the reference restoration design can achieve the objective of intertidal biodiversity rehabilitation. Indeed, because of the relatively slow accretion rates in the Northern basin (Fig. 6), the
restored tidal marsh still features the entire range of intertidal habitats after 50 years (Fig. 3).

The examples above illustrate the need to identify restoration design options that can steer rates of sediment accretion and vegetation development in line with restoration objectives. In this study, we focus on the impact of one specific design option: the inlet width (Fig. 2). Our model predicts that, for the setting studied here, higher differences between the two inlet widths lead to more contrasting sedimentation and vegetation patterns in the two basins (Fig. 9). This has two positive
outcomes. On the one hand, high accretion rates in the Southern basin bring fast vegetation development there, and potentially positive public perception for the restoration project. On the other hand, lower accretion rates in the Northern basin allow for long-term persistence of habitat diversity, which is an important objective in this project. Other important design options, such as the excavation of a channel network (Williams et al., 2002; Wallace et al., 2005; Hood, 2014), the manual planting of vegetation (O'Brien and Zedler, 2006; Staver et al., 2020), the infilling or lowering of areas, or the
creation of a landward slope, were beyond the scope of this study. However, new fundamental insight on the impact of such design options is crucial and should be investigated in the future. Our novel biogeomorphic model is made available for the scientific community in that perspective (see Data statement).

### 4.2 Evaluation of model performance

An important challenge for tidal marsh biogeomorphic models is that rates of sediment accretion and vegetation
development are highly uncertain (Wolters et al., 2005; French, 2006; Mossman at al., 2012; Spencer and Harvey, 2012). For example, exposure to wind waves and resulting sediment resuspension processes (Leonardi et al., 2016) can considerably lower accretion rates and potentially prevent any vegetation to establish (French et al., 2000). In contrast, improper inlet design can lead to very high accretion rates of poorly consolidated sediments that also limit vegetation establishment (Oosterlee et al., 2020). Other important sources of uncertainty for vegetation development include biotic control such as
benthic animals grazing on establishing plants (Paramor and Hughes, 2005; Silliman et al., 2005) and plant stress related to extreme events such as droughts or hurricanes (Howes et al., 2010). At this stage, our model is not equipped to evaluate these uncertainties.

However, to our knowledge, this paper presents the first application of a tidal marsh biogeomorphic model accounting for relevant fine-scale interactions (less than 1 m$^2$) between flow and stochastic, patchy vegetation establishment patterns, as
well as their long-term impact (several decades) at the landscape scale (several km$^2$) on vegetation and landform development. Previous studies were either limited to smaller domains (order of 1 km$^2$ or less – Temmerman et al., 2007; Best et al., 2018; Schwarz et al., 2018; Bij de Vaate et al., 2020; Wang et al., 2021), coarser grid resolutions (order of 100 m – Mariotti and Canestrelli, 2017), shorter simulation periods (order of 1 decade – Brückner et al., 2019), more simplified hydro-morphodynamics (Craft et al., 2009; Alizad et al., 2016; Spencer et al., 2016; Mariotti, 2020; Mariotti et al., 2020) or
more simplified vegetation dynamics (D'Alpaos et al., 2007; Belliard et al., 2015). However, our model does not include


certain processes that are accounted for in other recent marsh models, but that are considered not relevant for this specific study case, such as wind waves (Mariotti and Canestrelli, 2017; Best et al., 2018; Mariotti, 2020) and pond dynamics (Mariotti et al., 2020).

Studies that evaluate the performance of a tidal marsh biogeomorphic model against field observations of marsh
development over relevant spatio-temporal scales (several km² and decades) are rather scarce. In this paper, we show that our model results are in relatively good agreement with observations from tidal marshes close to the study site, in terms of sediment accretion rates on vegetated platforms (Fig. S3, supplementary material), vegetation cover development (Fig. S4, supplementary material) and channel geometric properties (Fig. 4). The discrepancies between model results and observations in channel width and channel depth, and to a lesser degree in channel cross-section area, may be related to
different factors. With a spatial resolution of 5 m, the model is unable to develop channels narrower than 10 to 20 m, while the observations come from remote sensing images with a spatial resolution of 50 cm, revealing channel widths as narrow as a few meters (Fig. 4c). Concomitantly, if the model overestimates the width of channels, this leads to a lower capacity for flow concentration and channel deepening through erosion, which can explain why the model predicts shallower channels, as compared to observations (Fig. 4d). Moreover, observations come from a much older marsh, which was also created by dike
breaching of a former polder area, but around 400 years ago (Jongepier et al., 2015). Fifty years after de-embankment, our restored tidal marsh is probably still at an earlier stage of development, as compared to the reference marsh. D'Alpaos et al. (2006) show that the width-to-depth ratio of channels decreases with marsh age. This is line with our model results, which indicate that channel depth increases over time, although channel width remains stable (Fig. S6, supplementary material), probably because of the grid resolution limitations discussed above. Finally, the SSC in the estuary is on average 1.5 times
higher in the vicinity of the study site, as compared to the adjacent reference marsh where observations come from (Sect. S2, supplementary material). However, our model results indicate that even reducing the sediment availability by a factor of 2 has nearly no impact on the channel geometric properties (Fig. S7, supplementary material). In any case, discrepancies in channel cross-section area are much smaller, as compared to channel width and channel depth, and they decrease with tidal prism (Fig. 4e). This suggests that appropriate volumes of water, and hence suspended sediments, are conveyed through the
channel networks and towards the intertidal platforms.

## 5    Conclusions

In this paper, we present a biogeomorphic model application to a specific tidal marsh restoration project by managed realignment at the Belgian/Dutch border along the macrotidal Scheldt Estuary. Our results indicate that the restored tidal marsh can keep pace with realistic rates of SLR and that its resilience is more sensitive to suspended sediment availability, in
agreement with current scientific knowledge. Our results further demonstrate that restoration design options can steer the biogeomorphic development of restored tidal marshes, and as such can be used as means to achieve specific restoration objectives.



Predicting the rate of restored tidal marsh development is very important, yet highly uncertain. Our novel biogeomorphic model, whose performance is here evaluated against observations in tidal systems close to the study site, is an attempt to reduce that uncertainty. Our promising results call for the evaluation of additional restoration design options.

**Code and data availability**

The biogeomorphic model Demeter is available at https://doi.org/10.5281/zenodo.5205258. The Python toolbox TidalGeoPro to compute geometric characteristics of tidal channel networks is available at https://doi.org/10.5281/zenodo.5205285. The multipurpose Python toolbox OGTools is available at https://doi.org/10.5281/zenodo.3994952. The data and source code to reproduce all figures and analyses are available at https://doi.org/10.5281/zenodo.5205261.

**Author contribution**

Conceptualization: OG, JvB, CS, TJB, JvdK, ST. Data curation: OG. Formal analysis: OG, JvB. Funding acquisition: OG, SF, TJB, JvdK, ST. Investigation: OG, JvB. Methodology: OG, JvB, CS, JPB, SF, TJB, JvdK, ST. Project administration: OG, SF, JvdK, ST. Resources: WV, JV. Software: OG. Supervision: SF, TJB, JvdK, ST. Validation: OG. Visualization: OG. Writing – original draft: OG. Writing – review and editing: all co-authors.

**Competing interests**

The authors declare that they have no conflict of interest.

**Acknowledgments**

This project has received funding from the Vlaams-Nederlandse Scheldecommissie (VNSC), the European Union's Horizon 2020 research and innovation program (Marie Skłodowska-Curie Actions – global postdoctoral fellowship – grant nr. 798222) and the Research Foundation – Flanders (FWO – fundamental research project – grant nr. G060018N). The resources and services used in this work were provided by the VSC (Flemish Supercomputer Center), funded by the Research Foundation – Flanders (FWO) and the Flemish Government. SF was partly funded by the USA National Science Foundation awards 1637630 (PIE LTER) and 1832221 (VCR LTER).



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
