# Peer review of "Biogeomorphic modeling to assess resilience of tidal marsh restoration to sea level rise and sediment supply"

_Earth Surface Dynamics, 2021_

## Author Response (AR1)

24 January 2022

Dear Associate Editor,

Dear Dr. Claire Masteller,

Please find below a detailed point-by-point response (black) to all referee comments (blue). Revised paragraphs of the manuscript are reproduced with tracked changes (new text, removed text) below our responses. Revised figures are also provided alongside their original versions for comparison.

We hope you appreciate our response to their constructive remarks, and that you will consider this revised version for publication in *Earth Surface Dynamics*. We want to sincerely thank you and the referees for helping to improve our manuscript.

Best regards,

Olivier Gourgue

**Referee #1**

This paper presents a biogeomorphic model applied to a specific tidal marsh restoration project in the Scheldt Estuary. The authors demonstrate model performance by way of application, comparing modeling results with morphological and ecological features of an active salt marsh located close to the restoration site. It is demonstrated that different options in the restoration scheme can critically lead to different evolutionary trajectories of the restored marsh, both in terms of morphological and ecological developments in space and time. The innovative side of the model lies in the fact that it combines different numerical techniques to couple both fine-scale vegetation dynamics and vegetation-flow interactions (occurring at sub metric scales) and the ecomorphodynamic evolution of the overall marsh systems (at km$^2$ scale). I have read the paper very carefully and found it much interesting and very well written. I only have minor comments that I'd like to submit to the authors before the paper can be published.

We thank the referee for their careful review of our manuscript, as well as for their positive and constructive remarks.

**Major comments**

l.230: It would be interesting to compare the values of SLR rates used here with the IPCC SLR projections for the same study area, to put the values used in this study in a proper context.

IPCC sea level rise rate projections for the period 2005-2055 at the estuary mouth range between 4.8 and 6.3 mm/yr. These are median projections for Representative Concentration Pathways 2.6 and 8.5 (IPCC, 2019). This is very much consistent with the reference value of 6 mm/yr used in our study. However, we also explain in the manuscript that this value of 6 mm/yr rather corresponds to the average rate of mean high-water level rise observed in the Scheldt Estuary over the last century, likely influenced by both global sea level rise and human-induced changes in the geomorphology of the estuary, such as dredging and dike construction. These local IPCC projections also illustrate that our two additional scenarios (i.e., 0 and 12 mm/yr) are rather extreme. We can therefore reasonably assume that our study covers the range of possible future sea level rise rates in the area.
* * *
Sect. 2.3.1 (1st paragraph)

We investigate the resilience of the restored tidal marsh to human-induced climate and environmental changes by considering different relative SLR rates and different SSC at the seaward boundary (Table 1). If our model can account for changes in MSL (Sect. 2.1), changes in MHWL are more relevant for the biogeomorphology of tidal marshes, as the intertidal elevation relative to MHWL determines the tidal inundation regime, hence affecting sediment accretion rates (Temmerman et al., 2004) and vegetation growth (Balke et al., 2016). Therefore, for the reference model scenario, we consider a SLR rate corresponding to the average rate of MHWL rise observed in the Scheldt Estuary over the last century, that is, 6 mm yr$^{-1}$ (Temmerman et al., 2004; Wang and Temmerman, 2013). This relatively high rate of MHWL rise is partly due to global

SLR but also likely amplified by local human-induced changes in the geomorphology of the estuary, such as historical embankment of intertidal areas and widening and deepening of the navigation channels (Smolders et al., 2015; Wang et al., 2019). We also consider two additional scenarios, with no (0 mm yr⁻¹) and high (12 mm yr⁻¹) SLR rate, respectively. In comparison, SLR rate projections for the period 2005-2055 at the estuary mouth range between 4.8 and 6.3 mm/yr (median projections for Representative Concentration Pathways 2.6 and 8.5 – IPCC, 2019). With these relatively extreme additional scenarios, we can therefore reasonably assume that we cover the range of possible future SLR rates in the area.

l.295: I am not entirely sure it is correct to refer to O'Brien's law here. The reason is twofold. First, the classic O'Brien's law is derived based on the tidal prism computed within tidal channels (not the overmarsh tidal prism as it was done here). Second, the exponent of the power-law relationship in O'Brien's law is well defined and typically equals ~6/7, which is quite different from the values proposed here (perhaps because of the difference in the way tidal prism is computed, as said before). Therefore, I'd rather refer to a generic tidal prism vs. cross-sectional channel area, without invoking O'Brien's law.

We agree that we should not refer to O'Brien's law in our analysis of channel cross-section surface area vs. overmarsh tidal prism. To remain consistent with the observations against which we compare our model results, we here follow the approach used by Vandenbruwaene et al. (2013, 2015) who argue that overmarsh tides (i.e., which overtop the mash platform level) are especially relevant in such analysis for tidal marsh channels, because maximum channel flow velocities typically occur when the surrounding platform is flooded and drained (Bayliss-Smith et al., 1979; Pethick, 1980; French and Stoddart, 1992). We have removed all references to O'Brien's law in the revised manuscript.

Sect. 2.4.3

[revised manuscript text omitted]

l.365: This is perhaps too big of a step since the propagation of suspended sediment depends not only on the tidal prism but also on well-known processes of sediment advection and dispersion. In fact, sediment transport of suspended sediment is by no means related to the tidal prism, the latter being only related to the channel cross-section as clearly demonstrated by the cited O'Brien's law, according to which the size (i.e., cross-section) of the channel depends on the flowing tidal prism regardless of the concentration sediments carried in suspension by tidal flows. This applies also to l.544-545.

We agree and have removed these sentences from the revised manuscript.

Sect. 3.2 (2nd paragraph)

The predicted channel network 50 years after de-embankment is slightly less dense, as compared with observations in a nearby tidal marsh (Fig. 4a), with mean unchanneled flow lengths of respectively 26.0 m (model) and 23.7 m (observations). The exponent of Hack's law is 0.908 for the model results and 0.909 for the observations (Fig. 4b). These values are statistically equal as the slopes of the linear regressions are not significantly different ($p = 0.913$ – Table S2, supplementary material). However, their intercepts are significantly different ($p = 0.007$), which means that the predicted channel lengths are smaller than those observed in the nearby natural marsh, with regards to the local watershed area. The exponent of the power law in Fig. 4e is 0.71 for the model results and 0.87 for the observations. These values are not statistically equal as the slopes of the linear regressions are slightly, but significantly different ($p = 0.023$). The intercepts are also significantly different ($p = 0.004$), which means that predicted channel cross-section areas are larger than those observed in the nearby tidal marsh, with regards to the local tidal prism. The relatively important deviations between model results and observations in terms of channel width and channel depth (Fig. 4c-d and Table S2, supplementary material) partly compensate each other, so that the discrepancy in channel cross-section area is much smaller, but also decreases with increasing tidal prism (Fig. 4e).

Sect. 4.2 (3rd paragraph)

Studies that evaluate the performance of a tidal marsh biogeomorphic model against field observations of marsh development over relevant spatio-temporal scales (several km² and decades) are rather scarce. In this paper, we show that our model results are in relatively good agreement with observations from tidal marshes close to the study site, in terms of sediment accretion rates on vegetated platforms (Fig. S4, supplementary material), vegetation cover development (Fig. S5, supplementary material) and channel geometric properties (Fig. 4). The discrepancies between model results and observations in channel width and channel depth, and to a lesser degree in channel cross-section area, may be related to different factors. With a spatial resolution of 5 m, the model is unable to develop channels narrower than 10 to 20 m, while the observations come from remote sensing images with a spatial resolution of 50 cm, revealing channel widths as narrow as a few meters (Fig. 4c). Concomitantly, if the model overestimates the width of channels, this leads to a lower capacity for flow concentration and channel deepening through erosion, which can explain why the model predicts shallower channels, as compared to observations (Fig. 4d). Moreover, observations come from a much older marsh, which was also created by dike breaching of a former polder area, but around 400 years ago (Jongepier et al., 2015). Fifty years after de-embankment, our restored tidal marsh is probably still at an earlier stage of development, as compared to the reference marsh. D'Alpaos et al. (2006) show that the width-to-depth ratio of channels decreases with marsh age. This is line with our model results, which indicate that channel depth increases over time, although channel width remains stable (Fig. S7, supplementary material), probably because of the grid resolution limitations discussed above. Finally, the SSC in the estuary is on average 1.5 times higher in the vicinity of the study site, as compared to the adjacent reference marsh where observations come from (Sect. S2, supplementary material). However, our model results indicate that even reducing

the sediment availability by a factor of 2 has nearly no impact on the channel geometric properties (Fig. S8, supplementary material). In any case, discrepancies in channel cross-section area are much smaller, as compared to channel width and channel depth, and they decrease with tidal prism (Fig. 4e).

I'd be curious to know model sensitivity to some of the input parameters, in particular those related to vegetation lateral expansion (e.g., $R_i^{exp}$). I think these parameters are critical in determining the evolution of marsh vegetation through time.

We have explored the model sensitivity to various vegetation input parameters, including the lateral expansion rate $R_i^{exp}$, but not in a systematic way for the present study. This is a very relevant topic, but we have already addressed it in a previous paper (Schwarz et al., 2018) and we further explore it in another paper in preparation. In general, fast colonizers (characterized by high number of establishing seedlings that produce homogeneous vegetation patterns) favor stabilization of pre-existing channels and consolidation of the landscape configuration, while slow colonizers (characterized by low number of establishing seedlings able to expand laterally, resulting in patchy vegetation patterns) facilitate the formation of new channels and thereby actively facilitates further landscape self-organization (Schwarz et al., 2018).

However, the scope of the present paper is on tidal marsh restoration and how different restoration design options can impact the biogeomorphic development of tidal marshes. That is why our model scenarios focus on real-life restoration design options (i.e., the size of the created tidal inlets), using fixed vegetation parameter values that are representative for the species present in the area, and that are well constrained for the Scheldt Estuary based on previous field studies (e.g., Silinski et al., 2016). Adding model scenarios with various vegetation parameter values would be an interesting theoretical model experiment, but not relevant for real-life marsh restoration, and hence would deviate from the scope of this paper. Nevertheless, for the sake of completeness, we have incorporated some examples as supplementary material of the revised manuscript, which illustrate that vegetation input parameters have rather limited impact on the long-term morphodynamics in the case studied here.

Sect. 2.3.3

In the reference model scenario, vegetation establishes randomly following different colonization strategies (i.e., either homogeneously with relatively high probability of establishment but no possibility to expand laterally, or patchily with relatively low probability of establishment but possibility to expand laterally to form growing patches – Sect. S1.2, supplementary material) in areas where environmental stressors allow for it (Sect. 2.1.2). This is the expected behavior supported by field observations for the three selected species representative for pioneer, middle and high marsh vegetation (Sect. S1.5.2, supplementary material). To illustrate the impact of the vegetation dynamics on the biogeomorphic feedbacks and the model results, we also consider  six variants of the reference model scenario (Table

Table S1 (new)

Table S1: Specifics of the reference model scenario variants used in Fig. S1 to S3.

| Variant name | Vegetation module parameterizations |
|---|---|
| No vegetation | No vegetation module. |
| Instantaneous colonization | Table S5. |
| Low establishment probability | Table S4, but $P_2^{est}$, $P_3^{est}$ and $P_{2,3}^{suc}$ divided by 10. |
| High establishment probability | Table S4, but $P_2^{est}$, $P_3^{est}$ and $P_{2,3}^{suc}$ multiplied by 10. |
| Low lateral expansion rate | Table S4, but $R_2^{exp}$, $R_3^{exp} = 1$ m/yr. |
| High lateral expansion rate | Table S4, but $R_2^{exp}$, $R_3^{exp} = 5$ m/yr. |

Sect. 3.1 (3rd paragraph)

Overall, the presence of vegetation has a positive impact on platform accretion rates in the Northern basin, although the speed of colonization has nearly no influence on the mean platform elevation 50 years after de-embankment (Fig. S1a, supplementary material). In the Southern basin, neither the presence of vegetation nor the speed of colonization seems to affect sediment accretion on the platforms (Fig. S1b, supplementary material), which suggests that the hydrodynamics is predominant in that part of the restored marsh. Locally, the vegetation dynamics can have remarkable geomorphic effects, such as the maintenance or disappearance of pre-excavated channels, whether we consider no vegetation, the reference vegetation dynamics, or instantaneous colonization (Fig. S2). In general, vegetation input parameters have a rather limited impact on the long-term morphodynamics (Fig. S3).

[Figure]

Fig. S3 (new)

Figure S3: Vegetation input parameter model scenarios (i.e., reference vegetation dynamics and four variants, respectively with low and high establishment probability (a, c), and with low and high lateral expansion rate (b, d) – Table S1). Evolution of the mean platform elevation with respect to the mean high-water level (MHWL) (a-b) and development of the vegetation cover (c-d).

Also, the authors state that different species have different $R_i^{exp}$, but looking at Table S4 it seems that $R_i^{exp}$ is held constant for all marsh perennials considered in this study. This would signify, if my interpretation is correct, that middle- and high-marsh species have nearly the same competitive ability, which I doubt is the case in real marshes.

The vegetation module is implemented such as each species can have a different mean expansion rate. In this study, the mean expansion rates for middle-marsh (*Scirpus maritimus*) and high-marsh (*Phragmites australis*) vegetation are determined based on remote sensing and literature data (see Table S3). This is pure coincidence if both species end up with the same value. However, that does not mean that both species have the same competitive ability, as the vegetation module simulates competitive interactions with a hierarchical model, where higher-rank species are stronger competitors able to outcompete lower-rank species (see Sect. S1.2 and Table S4). In our model, higher-rank species can displace lower-rank species, but not the other way around. Lower-rank species can only colonize after higher-rank species have died off. On the long term,

high-marsh vegetation (rank 3) will therefore always outcompete middle-marsh vegetation (rank 2) in its own niche.
* * *
Sect. S1.2 (new 2nd paragraph)

Our cellular automaton is implemented as a hierarchical model, where higher-rank species are stronger competitors able to outcompete lower-rank species. In our model, higher-rank species can displace lower-rank species, but not the other way around. Lower-rank species can only colonize after higher-rank species have died off. On the long term, high-rank species will therefore always outcompete lower-rank species in their own niche.
* * *
Also, related to this point, I wonder if the grid resolution for vegetation dynamics can be somehow dependent on the imposed $R_i^{exp}$ and numerical timestep (i.e., should the resolution not exceed a certain threshold for a given $R_i^{exp}$ and timestep to obtain reliable results with respect to vegetation dynamics)?

The grid resolution for vegetation dynamics is indeed dependent on the imposed expansion rate and numerical timestep. The number of iterations in the vegetation module (i.e., the ratio between the simulated period – one year – and the numerical time step) is determined as a function of the grid resolution and the mean expansion rate, by means of Eq. (S16) to (S19) (supplementary material).

**Minor comments**

l.15: add "restored" before "tidal marshes"

l.17: too generic. Explain why difficult to assess these key questions.

l.18: strange sentence…it looks like you're applying model by dike breaching.

l.19: add a comma after "transport"

l.24: it affects -> they affect (referred to options)

l.26: to more -> higher

l.26: diversity in terms of what? Morphological? Ecological?
* * *
Abstract

[revised manuscript text omitted]

**Referee #2**

The authors present an eco-geomorphological model with many interesting and novel features and apply the model to a design of a restoration project consisting of breaching of an existing dyke. The paper is very valuable as an application of state-of-the-art modelling to a specific restoration site with all the associated complications and uncertainties.

We thank the referee for their careful review of our manuscript, as well as for their positive and constructive remarks.

A critical feature of the model is that it can predict channel formation within the marsh as a result of the new hydrodynamic configuration due to the dyke breaches. It would be very useful to provide more detail on how the process of channel formation is implemented in the model. Is there a threshold value of shear stress? Is that a model parameter that is adjusted or calibrated? How does it compare to other sites/models?

Channels form through erosion and deposition of sediments, following Eq. (S5) to (S7). The critical shear stress for bed erosion is 0.5 N/m$^2$ for the fresh layer (i.e., sediments deposited during the simulation – Table S3) and 0.8 N/m$^2$ for the compacted layer (i.e., the sediment bed soil already present before marsh restoration – Table S3). This approach is consistent with a previous study on consolidation of accretional mudflats for the same tidal marsh restoration project (Zhou et al., 2016).
* * *
Sect. S1.5.1

The initial bed elevation is based on the project design (Sec. 2.2 and Fig. 2) and Lidar data before de-embankment. The bed is initially exclusively composed of a compacted layer. Tides are imposed into the system by defining water levels and flow velocities at the open boundary between the study site and the Scheldt Estuary, which is here approximately the isobath 5 m below the mean low water level. These boundary conditions are provided by a 3D hydrodynamic model of the estuary, which has been calibrated for a spring-neap cycle by comparison with measurements of water levels, flow velocities and water discharges (Maximova et al., 2014). To reduce the computational time, we do not simulate the entire range of tidal conditions of a full spring-neap cycle. Instead, we only select four different semi-diurnal tidal cycles from the estuarine model, which are representative of the standard range of tidal conditions that can be observed in that area. With high water levels of 2.05, 2.55, 2.87 and 3.25 m NAP, the selected tidal cycles have a frequency distribution of respectively 14.6%, 27.4%, 32.3% and 25.7%, as compared to historical measurements during the period 2007-2017. These frequency distributions are then used to determine the morphological acceleration factor $\alpha$ used for each semi-diurnal tidal cycle (Sec. 2.1). We simulate the impact of sea level rise by lowering the bed elevation every year by a value corresponding to the yearly increase of mean sea level. The suspended sediment concentration at the open boundary is constant and determined based on reported measurements (Vandenbruwaene et al., 2014; Sec. S2). All parameter values used in the hydro-morphodynamic module are based on previous studies in the same restored tidal

marsh area (Maximova et al., 2014; Zhou et al., 2016), the Scheldt Estuary (van Leussen, 1999; Van de Broek et al., 2018) and other intertidal environments (D'Alpaos et al., 2021). They are summarized in Table S3. The suspended sediment concentration at the open boundary and the rate of sea level rise vary according to model scenarios (Table 1).

Deposition of sediment and surface accretion is also quite important for the model results and there are a couple of points that would be good to have more information on. The first one is about the biological component of accretion, which includes the incorporation of plant litter into the soil (Morris et al., 2002) and that is not included in the model. It may well be that is not as important in this setting, but a comment on this would be valuable. For example, Breda et al. (2021) showed that the biological accretion can be of similar magnitude than the sediment related accretion. Those two accretion components may have a different behavior under climate change scenarios.

Sediment accretion in marshes of the Scheldt Estuary is dominated by the external supply by tides of suspended sediments, mostly of mineral nature, while organic matter only accounts for about 10% of the measured accretion rates (Temmerman et al., 2004). For this reason, our model does not explicitly simulate organic matter accretion locally produced by vegetation. However, organic matter accretion can be considered as implicitly compensated for by model calibration for total sediment accretion on vegetated platforms (Sect. 2.4.1). The calibration is indeed based on observed elevation changes, hence total accretion rates, including both mineral and organic contributions.

Sect. 2.2 (2nd paragraph)

Local environmental conditions are determinant for the development of restored ecosystems (Liu et al., 2021). The Scheldt Estuary, here defined as the tidal part of the Scheldt River, is a semidiurnal macrotidal estuary extending over 160 km. At a gauge station near Bath (Fig. 2b), the tidal range has been recorded to vary on average from 4.21 m at neap tides to 5.76 m at spring tides during the period 2011-2015, and the MHWL to rise at a rate of 5.7 mm yr$^{-1}$ during the period 1931-2004 (Wang and Temmerman, 2013). This MHWL rise rate is used here as proxy for SLR rate (Sect. 2.3). The study site lies in the brackish zone of the estuary, which is characterized by a steep gradient in salinity, with values ranging from 5 to 18 PSU (Van Damme et al., 2005; Meire et al., 2005). Therefore, only a limited number of vegetation species (Sect. 2.1) can cope with the local environmental conditions. Finally, the The local SSC is influenced by the presence of a maximum turbidity zone, where large volumes of cohesive sediments are concentrated and continually resuspended by the tidal flow (Baeyens et al., 1997; Chen et al., 2005; Meire et al., 2005). At the study site, the current average SSC is estimated at 63 mg l$^{-1}$ (Sect. S2, supplementary material). Sediment accretion in marshes of the Scheldt Estuary is dominated by the external supply by tides of suspended sediments, mostly of mineral nature, while organic matter only accounts for about 10% of the measured accretion rates (Temmerman et al., 2004). For this reason, our model does not explicitly simulate organic matter accretion locally produced by vegetation.

The other point is the formulation for deposition of fine sediment. Cohesive sediment deposition often involves the determination of a minimum depositional velocity below which fine particles (colloids) remain in suspension (Metha and McAnally, 2008). The model used in the paper does not have a minimum velocity threshold in its formulation, so it would be good to discuss the implications of such approach.

The existence of a minimum depositional velocity (or shear stress) below which fine particles remain in suspension is debated in the literature. In our model, we follow one of the well-established arguments that such threshold does not exist, and that it rather represents a threshold for erosion of freshly deposited sediments (Winterwerp, 2007). This approach agrees with field observations in the Chesapeake Bay (Sanford and Halka, 1993) and is often adopted in recent biogeomorphic models (e.g., Adams et al., 2016; Bryan et al., 2017; Mariotti, 2018; Zhang et al., 2019; Brückner et al., 2020).

Sect. S1.1 (3$^{rd}$ paragraph)

Sisyphe solves the depth-averaged advection-diffusion equation to simulate fluctuations of the depth-averaged suspended sediment concentration $C$:

Equation (S4)

where $E$ and $D$ are the rates of sediment erosion and deposition, respectively. The rate of sediment erosion is computed using the equation of Partheniades (1965):

Equation (S5)

where $M$ is the Partheniades constant and $\tau_e$ is the critical bed shear stress for sediment erosion. The rate of sediment deposition is computed using the equation of Einstein and Krone (1962):

Equation (S6)

where $w_s$ is the sediment settling velocity. The existence of a threshold shear stress below which sediments remain in suspension is debated in the literature. Here we follow one of the well-established arguments that such threshold does not exist, and that it rather represents a threshold for erosion of freshly deposited sediments (Winterwerp, 2007). This approach agrees with field observations in the Chesapeake Bay (Sanford and Halka, 1993) and is often adopted in recent biogeomorphic models (e.g., Adams et al., 2016; Bryan et al., 2017; Mariotti, 2018; Zhang et al., 2019; Brückner et al., 2020).

---

## Author Response (AR2)

31 March 2022

Dear Associate Editor,

Dear Dr. Claire Masteller,

Please find below a detailed point-by-point response (black) to your comments (blue). Revised paragraphs of the manuscript are reproduced with tracked changes (new text, removed text) below our responses.

We hope you appreciate our response to your constructive remarks, and that you will consider this revised version for publication in *Earth Surface Dynamics*. We want to sincerely thank you for helping to improve our manuscript.

Best regards,

Olivier Gourgue

**Associate Editor**

Thank you for your submission to *Earth Surface Dynamics*. Two reviews of the manuscript were generally positive and agreed that the results represent a novel contribution focused on the role of vegetation in modeling marsh morphodynamics. The reviewers asked for some clarifications throughout the manuscript that I feel have been sufficiently addressed.

Thank you for approving our responses to the referee comments.

I have gone through the revised manuscript and am suggesting some minor edits throughout to streamline, mostly focused on the introduction and methods.

Thank you for these additional suggestions. See our detailed point-by-point response below.

I would also encourage the authors to discuss the impact (or lack thereof) of the different vegetation scenarios on the results in the main text. I am aware that a more detailed discussion of this is included in the discussion, but because part of the novelty of the study is the application of the fine-scale vegetation model, I think the additional work done by the authors should be summarized in the main text.

With all due respect, we want to remind that the scope of the present paper is on tidal marsh restoration and how different restoration design options can impact the biogeomorphic development of tidal marshes. The impact of vegetation dynamics (through different vegetation scenarios) is an interesting theoretical model experiment that we have already addressed in a previous paper (Schwarz et al., 2018) and that we further explore in another paper in preparation.

Although it is not the main scope of the paper, we understand that the vegetation dynamics is an important aspect of the model. That is why we support our main study with variants of the reference scenarios to explore the model sensitivity to some aspects of the vegetation dynamics. In order not to distract the reader from the main scope of the paper, the technical details and the related results are presented as supplementary material (Table S1, Figures S1 to S3). However, these additional scenarios are also briefly presented in Sect 2.3.3 and the related results are summarized in Sect. 3.1 (3rd paragraph). Following your suggestion, we now also discuss them briefly in Sect. 4.2 (2nd paragraph).
* * *
Sect. 2.3.3

In the reference model scenario, vegetation establishes randomly following different colonization strategies (i.e., either homogeneously with relatively high probability of establishment but no possibility to expand laterally, or patchily with relatively low probability of establishment but possibility to expand laterally to form growing patches – Sect. S1.2, supplementary material) in areas where environmental stressors allow for it (Sect. 2.1.2). This is the expected behavior supported by field observations for the three selected species representative for pioneer, middle and high marsh vegetation (Sect. S1.5.2, supplementary material). To illustrate the impact of the vegetation dynamics on the biogeomorphic feedbacks

and the model results, we also consider six variants of the reference model scenario (Table S1, supplementary material).

Sect. 3.1 (3ʳᵈ paragraph)

Overall, the presence of vegetation slightly increases the rate of platform accretion in the Northern basin, although the speed of colonization has nearly no influence on the mean platform elevation 50 years after de-embankment (Fig. S1a, supplementary material). In the Southern basin, neither the presence of vegetation nor the speed of colonization seems to affect sediment accretion on the platforms (Fig. S1b, supplementary material), which suggests that the hydrodynamics is predominant in that part of the restored marsh. Locally, the vegetation dynamics can have remarkable geomorphic effects, such as the maintenance or disappearance of pre-excavated channels, whether we consider no vegetation, the reference vegetation dynamics, or instantaneous colonization (Fig. S2, supplementary material). In general, vegetation input parameters have a rather limited impact on the long-term morphodynamics (Fig. S3, supplementary material).

Sect. 4.2 (2ⁿᵈ paragraph)

However, to our knowledge, this paper presents the first application of a tidal marsh biogeomorphic model accounting for relevant fine-scale interactions (less than 1 m$^2$) between flow and stochastic, patchy vegetation establishment patterns, as well as their long-term impact (several decades) at the landscape scale (several km$^2$) on vegetation and landform development. Previous studies were either limited to smaller domains (order of 1 km$^2$ or less – Temmerman et al., 2007; Best et al., 2018; Schwarz et al., 2018; Bij de Vaate et al., 2020; Wang et al., 2021), coarser grid resolutions (order of 100 m – Mariotti and Canestrelli, 2017), shorter simulation periods (order of 1 decade – Brückner et al., 2019), more simplified hydro-morphodynamics (Craft et al., 2009; Alizad et al., 2016; Spencer et al., 2016; Mariotti, 2020; Mariotti et al., 2020) or more simplified vegetation dynamics (D'Alpaos et al., 2007; Belliard et al., 2015). However, our model does not include certain processes that are accounted for in other recent marsh models, but that are considered not relevant for this specific study case, such as wind waves (Mariotti and Canestrelli, 2017; Best et al., 2018; Mariotti, 2020) and pond dynamics (Mariotti et al., 2020). While previous studies showed that vegetation dynamics can considerably impact tidal channel morphodynamics (Schwarz et al., 2018; Temmerman et al., 2007), our model results suggest that it is not a dominant process in the case studied here. Locally, the vegetation dynamics can affect the sustainability of certain channels in the Northern basin (Fig. S2, supplementary material), but overall, sediment accretion on the platforms is much more sensitive to hydrodynamic processes such as SLR and sediment supply (Fig. 6) than to vegetation dynamics (Fig. S1 and S3, supplementary material). This calls for further research on the environmental conditions under which the vegetation dynamics can be more impactful on the morphodynamics (e.g., lower tidal range).

Because these are mainly comments on the text, not on the methods or analysis, I am marking these as minor revisions.

**Minor edits**

Line 58: Replace "while" with "then".

Line 67: Replace "develop" with "continue to develop over time".

Lines 67-70: Replace "For example, several studies point at many restored sites that, in comparison with natural tidal marshes, underperform in terms of (…)" with "For example, several studies indicate that restored sites underperform in term of (…) when compared to their natural counterparts".
* * *
Sect. 1 (2nd paragraph)

Managed realignment, which consists in shifting the line of coastal defense structures landward of their existing position, can create space for tidal marsh restoration or creation.  This practice has grown in popularity over the last two decades (French, 2006; Turner et al., 2007), especially in the context of coastal squeeze and landward movement of the mean low water mark due to SLR and storms (Doody, 2013). Practically, a second line of defense is built landwards,  then the first one is breached. The number and size of breaches are important design choices (Hood, 2014, 2015) and vary greatly between projects (e.g., Friess et al., 2014; Dale et al., 2017). As breaches become the inlets of the restored marshes, they have an important control on water and sediment volumes entering and leaving the system during each tidal cycle, and hence on sediment accretion rates (Oosterlee et al., 2020). Other important design measures may involve excavating an initial channel network and treating soil conditions to facilitate soil drainage (O'Brien and Zedler, 2006), planting manually vegetation tussocks to ensure vegetation encroachment (Staver et al., 2020), or building hydraulic structures to control the tidal range and create optimal ecological conditions for vegetation development (Maris et al., 2007; Oosterlee et al., 2018). These design choices are mainly driven by restoration objectives and local environmental conditions. Yet, there is high uncertainty in how restored tidal marshes continue to develop over time. For example, several studies  indicate that restored sites  underperform in terms of biodiversity (Wolters et al., 2005; Mossman et al., 2012), topographic diversity (Lawrence et al., 2018), groundwater dynamics (Tempest et al., 2015; Van Putte et al., 2020) and biogeochemical functioning, including carbon sequestration (Santín et al., 2009; Suir et al., 2019) when compared to their natural counterparts. These outcomes can potentially hamper marsh ecosystem functions and the initial restoration objectives.
* * *
Line 72: Remove "yet so important".

Line 72: Replace "for example" with "in some cases".

Line 77: Remove "opinion".

Line 83: Remove "that are based on state-of-the-art scientific knowledge, and".

Line 83: Replace "allow" with "are able".

Sect. 1 (3rd paragraph)

The rate at which tidal marshes develop in restoration projects is highly uncertain, yet so important. For example In some cases, sediment accretion rates determine whether restored tidal marshes can keep pace with local rates of SLR (Kirwan et al., 2010; Vandenbruwaene et al., 2011a; Webb et al., 2013; Kirwan et al., 2016). The establishment rate of pioneer vegetation and the succession towards climax vegetation may depend on small windows of opportunity that are very difficult to predict (Chambers et al., 2003; Hu et al., 2015; Cao et al., 2018). Furthermore, the rate of development is at the center of the tension between public perception and restoration objectives. The public opinion is often very critical towards marsh restoration by managed realignment, as it implies the loss of valuable land, laboriously reclaimed by previous generations (Temmerman et al., 2013). On the one hand, fast development allows to quickly reach target habitats, which may support a positive public perception, but involves the risk of fast development towards a monotone climax ecosystem state. On the other hand, slow development (e.g., including bare mudflats) increases the risk of negative public perception in the first years, but may lead to long-term persistence of high habitat diversity with different stages of succession. All these examples illustrate the need for modeling tools that are based on state-of-the-art scientific knowledge, and that allow to can predict how fast restored tidal marshes develop and how development rates can be steered by restoration design.

Line 208: Unclear what "(Sect. 2.1)" is referencing. I'd suggest adding the vegetation species by name here.

Sect. 2.2 (2nd paragraph)

Local environmental conditions are determinant for the development of restored ecosystems (Liu et al., 2021). The Scheldt Estuary, here defined as the tidal part of the Scheldt River, is a semidiurnal macrotidal estuary extending over 160 km. At a gauge station near Bath (Fig. 2b), the tidal range has been recorded to vary on average from 4.21 m at neap tides to 5.76 m at spring tides during the period 2011-2015, and the MHWL to rise at a rate of 5.7 mm yr-1 during the period 1931-2004 (Wang and Temmerman, 2013). This MHWL rise rate is used here as proxy for SLR rate (Sect. 2.3). The study site lies in the brackish zone of the estuary, which is characterized by a steep gradient in salinity, with values ranging from 5 to 18 PSU (Van Damme et al., 2005; Meire et al., 2005). Therefore, only a limited number of vegetation species (Sect. 2.1 e.g., *Aster tripolium, Scirpus maritimus* and *Phragmites australis*) can cope with the local environmental conditions. The local SSC is influenced by the presence of a maximum turbidity zone, where large volumes of cohesive sediments are concentrated and continually resuspended by the tidal flow (Baeyens et al., 1997; Chen et al., 2005; Meire et al., 2005). At the study site, the current average SSC is estimated at 63 mg l-1 (Sect. S2, supplementary material). Sediment accretion in marshes of the Scheldt Estuary is dominated by the external supply by tides of suspended sediments, mostly of mineral nature, while organic matter only accounts for about 10% of the measured

accretion rates (Temmerman et al., 2004). For this reason, our model does not explicitly simulate organic matter accretion locally produced by vegetation.

Lines 256-259: This seems like an important detail. I suggest taking it out of parentheses and just making it its own sentence.

Sect. 2.3.3

In the reference model scenario, vegetation establishes randomly following different colonization strategies (i.e., either homogeneously with relatively high probability of establishment but no possibility to expand laterally, or patchily with relatively low probability of establishment but possibility to expand laterally to form growing patches – Sect. S1.2, supplementary material) in areas where environmental stressors allow for it. Pioneer marsh vegetation establishes homogeneously with a relatively high probability of establishment but with no possibility to expand laterally. Middle and high marsh vegetation establish patchily with a relatively low probability of establishment but with the possibility to expand laterally to form growing patches (Sect. 2.1.2). This is the expected behavior supported by field observations for the three selected species representative for pioneer, middle and high marsh vegetation (Sect. S1.5.2, supplementary material). To illustrate the impact of the vegetation dynamics on the biogeomorphic feedbacks and the model results, we also consider six variants of the reference model scenario (Table S1, supplementary material).

Line 262: Are these different form the models in Table 1? What is the basis for those models only appearing in the supplement and not the main text? Particularly if one of the key advances of this study is to include the vegetation feedbacks in the modeling study.

As mentioned above, the scope of the paper is on tidal marsh restoration and how different restoration design options can impact the biogeomorphic development of tidal marshes. This is investigated with the scenarios presented in Table 1. To support our model results, we also explored the model sensitivity to some aspects of the vegetation dynamics with additional scenarios presented in Table S1. In order not to distract the reader from the main scope of the paper, this table, which contains rather technical details, is presented as supplementary material.

Line 339: What do you define as positive impact? Add a sentence to be more specific here.

Sect. 3.1 (3rd paragraph)

Overall, the presence of vegetation has a positive impact on slightly increases the rate of platform accretion rates in the Northern basin, although the speed of colonization has nearly no influence on the mean platform elevation 50 years after de-embankment (Fig. S1a, supplementary material). In the Southern basin, neither the presence of vegetation nor the speed of colonization seems to affect sediment accretion on the platforms (Fig. S1b, supplementary material), which suggests that the hydrodynamics is predominant in that part of the restored marsh. Locally, the vegetation dynamics can have remarkable geomorphic effects, such as the maintenance or disappearance of pre-excavated channels, whether we consider no vegetation, the reference

vegetation dynamics, or instantaneous colonization (Fig. S2, supplementary material). In general, vegetation input parameters have a rather limited impact on the long-term morphodynamics (Fig. S3, supplementary material).

Lines 345-346: It is difficult to evaluate just how much the vegetation is having an effect, particularly when it looks like the bulk of the deposition is occurring when the vegetation has yet to colonize or is still very sparse.

With all due respect, we believe that this is what this paragraph is about. In the case studied here, because of the local environmental conditions (e.g., high tidal range), the morphodynamics does not seem to be very sensitive to the vegetation dynamics. We have added a paragraph in the discussion to clarify it.

Sect. 4.2 (2$^{nd}$ paragraph)

However, to our knowledge, this paper presents the first application of a tidal marsh biogeomorphic model accounting for relevant fine-scale interactions (less than 1 m$^2$) between flow and stochastic, patchy vegetation establishment patterns, as well as their long-term impact (several decades) at the landscape scale (several km$^2$) on vegetation and landform development. Previous studies were either limited to smaller domains (order of 1 km$^2$ or less – Temmerman et al., 2007; Best et al., 2018; Schwarz et al., 2018; Bij de Vaate et al., 2020; Wang et al., 2021), coarser grid resolutions (order of 100 m – Mariotti and Canestrelli, 2017), shorter simulation periods (order of 1 decade – Brückner et al., 2019), more simplified hydro-morphodynamics (Craft et al., 2009; Alizad et al., 2016; Spencer et al., 2016; Mariotti, 2020; Mariotti et al., 2020) or more simplified vegetation dynamics (D'Alpaos et al., 2007; Belliard et al., 2015). However, our model does not include certain processes that are accounted for in other recent marsh models, but that are considered not relevant for this specific study case, such as wind waves (Mariotti and Canestrelli, 2017; Best et al., 2018; Mariotti, 2020) and pond dynamics (Mariotti et al., 2020). While previous studies showed that vegetation dynamics can considerably impact tidal channel morphodynamics (Schwarz et al., 2018; Temmerman et al., 2007), our model results suggest that it is not a dominant process in the case studied here. Locally, the vegetation dynamics can affect the sustainability of certain channels in the Northern basin (Fig. S2, supplementary material), but overall, sediment accretion on the platforms is much more sensitive to hydrodynamic processes such as SLR and sediment supply (Fig. 6) than to vegetation dynamics (Fig. S1 and S3, supplementary material). This calls for further research on the environmental conditions under which the vegetation dynamics can be more impactful on the morphodynamics (e.g., lower tidal range).

Given that this is one of the stated novelties of the paper, some explicit evaluation of this would be helpful to clarify this point for readers. I recognize that this information is in the supplement, but some of those findings can be briefly summarized in the main text. There are sections for the impact of SSC and the impact of inlet design, so the authors may consider also adding a short note about the vegetation impacts.

As mentioned above, the scope of the paper is on tidal marsh restoration and how different restoration design options can impact the biogeomorphic development of tidal marshes. The additional scenarios to explore the model sensitivity to some aspects of the vegetation dynamics are presented in Sect 2.3.3 and the related results are summarized in Sect. 3.1 (3$^{rd}$ paragraph) and now discussed in Sect. 4.2 (2$^{nd}$ paragraph). However, in order not to distract the reader from the main scope of the paper, we decided to have no specific section on the impact of vegetation dynamics in the results and discussion sections, and to present the related figures as supplementary material.
* * *
Sect. 2.3.3

[revised manuscript text omitted]